# Agent Forecasting at Flexible Horizons using ODE Flows

**Alexander Radovic** [1]    **Jiawei He** [1]    **Janahan Ramanan** [1]    **Marcus A. Brubaker** [1 2 3]    **Andreas M. Lehrmann** [1]

## Abstract

In this work we describe OMEN, a neural ODE based normalizing flow for the prediction of marginal distributions at flexible evaluation horizons, and apply it to agent position forecasting. OMEN's architecture embeds an assumption that marginal distributions of a given agent moving forward in time are related, allowing for an efficient representation of marginal distributions through time and allowing for reliable interpolation between prediction horizons seen in training. Experiments on a popular agent forecasting dataset demonstrate significant improvements over most baseline approaches, and comparable performance to the state of the art while providing the new functionality of reliable interpolation of predicted marginal distributions between prediction horizons as demonstrated with synthetic data.

## 1. Introduction

Autonomous driving has benefited tremendously from recent progress in deep learning and computer vision (Grigorescu et al., 2019). The capability of recognizing traffic signs (Arcos-García et al., 2018; Zhou et al., 2020), localizing pedestrians (Mao et al., 2017; Liu et al., 2019), etc. makes it possible for autonomous vehicles to "see" the world (Zhao et al., 2019). However, one critical component for safe and efficient planning in autonomous vehicles is an accurate prediction of the future position of such agents (such as pedestrians or moving vehicles) in the environment (Mozaffari et al., 2019; Rudenko et al., 2020). Despite the importance of the position prediction problem, the performance on this task is still far from satisfactory because of the following challenging requirements: (1) predictions must be conditioned on the environment, as contextual clues are

essential for an accurate prediction (an example is given in Fig. 1a); and (2) predictions are required to be highly multi-modal (shown in Fig. 1b) as the real-world environment often exhibits junctions where an agent has several distinct possible future trajectories, and mode collapse in these moments could lead to disastrous planning outcomes.

It is common to frame the agent forecasting task as learning marginal distributions over potential agent positions (Makansi et al., 2019; Oh & Valois, 2019; Zieba et al., 2020), also known as "occupancy maps", a popular representation in planning for robotics and autonomous vehicles (Grigorescu et al., 2019; Mozaffari et al., 2019). By predicting the marginal distribution at a specific point in time, these methods are often superior at capturing the complex multi-modal nature of the data, avoiding the challenges of generating diverse trajectories (Ma et al., 2020). In addition, while the underlying process of an agent's trajectory is continuous, most popular forecasting models operate on a discretized representation of time chosen during training (Whittle, 1951; Rhinehart et al., 2018; Mozaffari et al., 2019; Makansi et al., 2019; Salinas et al., 2019; Tang & Salakhutdinov, 2019; Rhinehart et al., 2019; Oh & Valois, 2019; Zieba et al., 2020). The granularity of time-steps used in training constrains the resolution and utility of these approaches. Please refer to the Appendix for a detailed discussion on related works.

Recently, Deng et al. (2020) demonstrated a conditional temporal process which can produce marginals and trajectories fully continuous in time. However, the expressiveness of this approach is ultimately bounded by the formulation as a stochastic process, taken in their paper to be a differential deformation of the Wiener process.

Building upon this approach, we propose a novel normalizing flow based architecture motivated by the assumption of modelling a continuous temporal process, where our model defines a new temporal process rather than deforming an existing one. The described model is shown in Fig. 1c. The main contributions of this work are summarized as following: (1) An expressive, multi-modal conditional normalizing flow based model for predicting agent positions. (2) A model capable of predicting at flexible horizons, including those not seen in training. (3) A flow architecture that embeds assumptions that, for a continuous process, pre-

---

[*]Equal contribution  [1]Borealis AI, Toronto, Ontario, Canada [2]York University, Toronto, Ontario, Canada [3]Vector Institute, Toronto, Ontario, Canada. Correspondence to: Alexander Radovic <alex.radovic@borealisai.com>.

Third workshop on *Invertible Neural Networks, Normalizing Flows, and Explicit Likelihood Models* (ICML 2021). Copyright 2021 by the author(s).

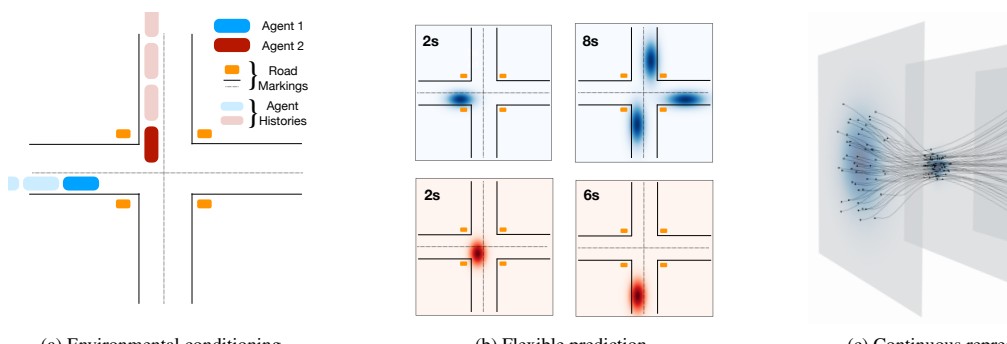

(a) Environmental conditioning.    (b) Flexible prediction.    (c) Continuous representation.

*Figure 1.* **Overview. (a)** Environmental conditioning. Agent location prediction requires synthesis of complex conditioning information, e.g. road markings, agent histories, lidar, video data. **(b)** Flexible prediction. Our goal is to predict marginals across agent locations at any choice of time, shown here for agent 1 (top, blue) and agent 2 (bottom, red). **(c)** Continuous representation. We propose a continuous flow based architecture, explicitly connecting marginal predictions across horizons. Here a base distribution (left) is connected to a marginal prediction at 2 seconds (middle) and 8 seconds (right) by a single neural ODE. Black lines show sample trajectories, corresponding to solutions to the ODE with an initial value taken from the base distribution.

dicted marginal distributions deform smoothly in time. (4) Demonstrations on both synthetic data, and an important agent forecasting dataset.

## 2. Method

In this section, we present our model and its optimization. We consider the task of predicting marginal distributions over future vehicle positions based on asynchronous conditioning information. Specifically, given 2D positional data $\mathbf{x} := \{\mathbf{x}_{(t_i')}\}_i$ for an agent at asynchronous times $t_i' \in T'$, we are interested in the marginal distributions $p(\{\mathbf{x}_{(t_i)}\}_i)$, with $T \ni t_i > \max(T')$, where $T$ is a set of target horizons. In practice, we may also have image-based auxiliary information $\mathbf{a} = \{\mathbf{a}_{(t_i')}\}_i$, such as Lidar scans, and write $\phi := \{\mathbf{x}, \mathbf{a}\}$ to summarize all available information up to time $t_0 := \max(T')$. Due to the nature of the data we work with, we will principally refer to timepoints (e.g., $t_i$, $t_i'$), however, our model is continuous in time, and as such it will at times be necessary to refer to the continuous axis of time $t$ which those observations lie on. Further the positional data $\mathbf{x}_{(t_i)}$ is taken to be the discrete vectorized observations of a function $x(t)$.

Our approach builds upon previous work on normalizing flows (NFs) and its continuous counterparts. We refer the reader to (Rezende & Mohamed, 2016; Chen et al., 2018; Grathwohl et al., 2018; Papamakarios et al., 2019; Kobyzev et al., 2020) for additional details.

### 2.1. Normalizing Flows with Informative Base Distributions

In the normalizing flow literature, it is usually assumed that a sufficiently expressive flow makes the choice of base distribution irrelevant (Papamakarios et al., 2019; Kobyzev

et al., 2020), and is therefore commonly chosen as a simple Gaussian distribution. However, recent works (Deng et al., 2020; Jaini et al., 2020; Mahajan et al., 2020) have started exploring constructions where the choice of base distribution embeds information about the target distribution, allowing good approximations of the target distribution with simpler flow transforms. For example, Jaini et al. (2020) demonstrated that for a target distribution with heavy tails, choosing a base distribution with similar heavy tails can be more effective than a wide variety of modern complex NF transforms in capturing the target distribution accurately.

Inspired by the aforementioned discussion, we suggest that to model the distribution of $p_t(x(t)|\phi)$ for a range of values of $t > t_0$, a desired property of the model would be that the distributions of $p_t(x(t)|\phi)$ and $p_{t+\epsilon}(x(t+\epsilon)|\phi)$ should be similar for small $\epsilon$ and identical when $\epsilon \to 0$.[1] In other words, $p_t(x(t))$ can be served as an *informative* base distribution of $p_{t+\epsilon}(x(t+\epsilon))$. This can be realized by incrementally transforming distributions as time progresses. Therefore, we can formulate the proposed model as follows: at any target time in the future, we can describe the target distribution $p_{t+\epsilon}(x(t+\epsilon))$ as a transform $f$ (taken to be a normalizing flow) forward in time from the previous time-step $p_t(x(t))$,

$$ p_{t+\epsilon}(x(t+\epsilon)) = p_t(f^{-1}(x(t+\epsilon))) \left| \det \frac{\partial f^{-1}}{\partial x(t+\epsilon)} \right|. \quad (1) $$

In addition, we can take advantage of the fact that the series of flow transforms at any point in a sequence building out from the base distribution represents a valid normalizing flow. Therefore, we can implement a network with multiple outputs, with each output further from the base distribution

---

[1]To ease notation, we drop references to the conditioning information $\phi$ from now on.

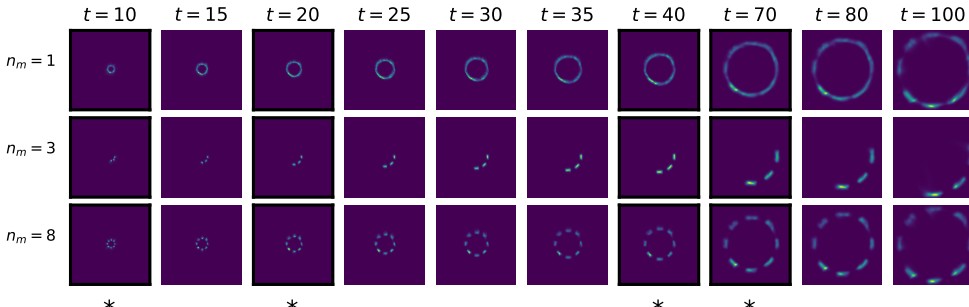

*Figure 2.* **Interpolation in Time with Synthetic Data.** Plots of predicted likelihood vs. $x-$ and $y$-coordinates at a series of times into the future. The number of modes $n_m$ was provided as conditioning information, and times marked with * were seen in training. The times shown here are a subset of those in Table 1.

learning to predict a point further into the future. This formulation, inspired by recent progress on informative base distributions for NFs (Deng et al., 2020; Jaini et al., 2020; Mahajan et al., 2020), motivates our proposed architecture.

### 2.2. Representation Through a Continuous Conditional Normalizing Flow

Built upon the discrete model described above, we realise the proposed NF architecture by adopting a neural ODE representation. With this approach, we find our model can, with minimal regularization (Finlay et al., 2020), learn reasonable interpolations between evaluation points at training phase, allowing us to produce valid marginal distributions at arbitrary target times. The proposed model utilizes the above-discussed "prior" intuition when constructing marginal distributions by taking marginals at earlier time-steps as informative base distributions. A illustration outlining this approach is available in the Appendix.

To facilitate asynchronous conditioning when predicting conditional marginal distributions, a vector of conditioning information from an encoder model is passed to the neural ODE. Specifically, as an extension to (Chen et al., 2018; Grathwohl et al., 2018), this information is concatenated to the input of a fully-connected neural network $f$ described by the neural ODE transform $\frac{\partial z(t)}{\partial t}$, such that for some parameters $\theta$ and conditioning information $\phi$ we have

$$f(z(t), t, \phi; \theta) = \frac{\partial z(t)}{\partial t}. \quad (2)$$

Following (Chen et al., 2018; Grathwohl et al., 2018), setting $z(t_i)$ to match the an observation $x_{(t_i)}$, we can solve the initial value problem to find the equivalent point in the base distribution $z(0)$:

$$\log p(z(t_i)) = \log p(z(0)) - \int_0^{t_i} \text{tr} \frac{\partial f}{\partial z(t)} dt. \quad (3)$$

Calculating likelihood estimates at multiple horizons of interest simply requires solving the initial value problem

for each different choice of $t$, where here the temporal axis of the ODE is explicitly aligned with the axis of time in the dataset of interest. A 'trajectory' can be generated by first sampling from the base distribution and then solving the ODE for the sampled point at $t = 0$, however unlike a true trajectory the only source of stochasticity is the initial sample from the base distribution.

**Training.** The proposed model can be optimized by minimizing the mean negative log-likelihood of distributions at $|T|$ target horizons. Therefore, our optimization objective can be formulated as:

$$\mathcal{L}_{\text{NLL}}(f(z(t), t, \phi; \theta), \{\mathbf{x}_{(t_i)}\}_i) = -\sum_{i=0}^{|T|} \log(p_{t_i}(\mathbf{x}_{(t_i)}|\phi, t_i, \theta))). \quad (4)$$

Note that although the model is trained on a finite selection of time-steps, inference (evaluation) can be conducted at any time.

## 3. Evaluation

In this section we demonstrate the ability of the model to generate realistic position estimates for an agent at a future time in both synthetic datasets and a complex autonomous driving environment.

### 3.1. Position Estimation on Synthetic 2D Data

In order to explore our model's ability to interpolate and extrapolate through time we created a synthetic multi-modal temporal process dataset. This process consists of radially growing angular distribution bands. The bands have 3 different modes. The modes control the angular division of distributional bands. At each time-step the radial distance of the band grows with step length drawn from a normal distribution. Conditioning information on the number of modes $n_m \in \{1, 3, 8\}$ is encoded using an MLP before concatenated to every layer of the neural ODE flow in place of $\phi$. Our model was trained on a specific subset of time

| $n_m$ | | Prediction Horizon | | | | | | | | | | | | |
|---|---|---|---|---|---|---|---|---|---|---|---|---|---|---|
| | | 10* | 15 | 20* | 25 | 30 | 35 | 40* | 50* | 60* | 70* | 80 | 90 | 100 |
| 1 | ● | 1.588 | 2.159 | 2.399 | 2.713 | 2.938 | 3.117 | 3.335 | 3.656 | 3.942 | 4.195 | 4.441 | 4.78 | 5.426 |
| | ○ | 1.834 | 1.957 | 2.311 | 2.565 | 2.782 | 3.012 | 3.227 | 3.579 | 3.931 | 4.112 | 4.443 | 4.578 | 4.770 |
| 3 | ● | -0.006 | 0.344 | 0.676 | 0.974 | 1.188 | 1.379 | 1.580 | 1.930 | 2.181 | 2.435 | 2.677 | 2.942 | 3.1551 |
| | ○ | 0.719 | 0.649 | 0.779 | 0.994 | 1.248 | 1.476 | 1.641 | 1.956 | 2.239 | 2.39 | 2.670 | 2.915 | 3.231 |
| 8 | ● | 1.092 | 1.516 | 1.805 | 2.153 | 2.380 | 2.572 | 2.788 | 3.064 | 3.321 | 3.558 | 3.803 | 4.150 | 4.601 |
| | ○ | 1.681 | 1.726 | 1.91 | 2.133 | 2.368 | 2.562 | 2.71 | 3.082 | 3.348 | 3.662 | 3.842 | 4.092 | 4.329 |

*Table 1.* **Performance (NLL) on Target Horizons.** The number of modes $n_m$ is treated as a conditioning variable of the model. ● marks the model trained on times marked with * for respective columns, and interpolated/extrapolated to times with no *. ○ marks a model trained and evaluated only on times not marked with a *. Performance can be seen to be broadly equivalent between the two models, demonstrating an ability to both interpolate and extrapolate to times unseen in training.

| Method | Test $\hat{e}$ |
|---|---|
| PRECOG-ESP (Rhinehart et al., 2019) | $0.634 \pm 0.006$ |
| HCNAF (Oh & Valois, 2019) | 0.114 |
| CTFP* (Deng et al., 2020) | $0.500 \pm 0.014$ |
| OMEN* | $0.185 \pm 0.002$ |
| OMEN-discrete | $0.144 \pm 0.006$ |
| OMEN-nocon* | $0.791 \pm 0.010$ |

*Table 2.* PRECOG-Carla single agent forecasting evaluation. Lower is better. All models use PRECOG-Carla Town 1 Training set in training, and are evaluated on the PRECOG-Carla Town 1 test set. OMEN, OMEN-nocon, and CTFP, marked with *, are able to produce likelihood estimates for unseen target horizons.

points $t \in \{10, 20, 40, 50, 60, 70\}$, then evaluated at a variety times never seen in training, including examples of both interpolation and extrapolation. Performance on log-likelihood estimation are comparable to a model trained explicitly on held out times. Full results are show in Table 1, qualitative results are shown in Fig. 2.

### 3.2. Agent Forecasting Experiments

**Baselines and Ablations.** Results from our model are compared to several leading approaches for likelihood estimation on agent forecasting. Minor modifications to the CTFP model (Deng et al., 2020), a discrete ablation of OMEN, and an ablation of OMEN without conditioning information are described in the appendix. While all baselines are capable of producing likelihood estimates for times seen in training, only the full OMEN model, its ablation without conditioning information, and the CTFP model (Deng et al., 2020) are able to produce likelihood estimates for unseen time points.

**Metrics.** Following Rhinehart et al. (2019), results are presented here using the extra nats metric $\hat{e}$, which provides a normalized and bounded likelihood metric, $\hat{e} := H(p', q) - H(\eta)/(|T| \cdot N_D)$, where $H(p', q)$ is the cross-entropy between the true distribution $p'$ perturbed by some noise $\eta$ (taken here as $\eta = \mathcal{N}(\mathbf{0}, 0.01^2 \cdot \mathbf{I}$ to match Rhine-

hart et al. (2019)) and our models prediction $q$, $N_D$ is the number of dimensions in the position data, and $H(\eta)$ can be calculated analytically. Following Oh & Valois (2019) we combine our marginal predictions at separate horizons to form a joint prediction to allow direct comparison to Rhinehart et al. (2019).

**PRECOG Carla Dataset.** The PRECOG Carla dataset (Rhinehart et al., 2019) is comprised of the complex simulated trajectories of an *autopilot* and four other agents in the Carla traffic simulation (Dosovitskiy et al., 2017), and includes additional Lidar data centred on the main autopilot agent. Here train, validation, and test data subsets were chosen to match Rhinehart et al. (2019). OMEN and its ablations were trained to minimize the NLL of PRECOG Carla's autopilot for all future time-steps available in the dataset. Results are presented in Table 2, and plots showing example predictions are available in the Appendix. We also refer the readers to the Appendix for further implementation details.

## 4. Conclusion

We presented a normalizing flow based architecture with a structure motivated by the assumption of modelling a continuous temporal process. Experimental evidence suggested that the constraints that allow for the smooth interpolation of likelihood estimates did cause some degradation in performance, however novel new capabilities are demonstrated in comparison to other leading approaches for likelihood estimation on agent forecasting. Specifically we demonstrated the ability to conditionally model complex processes, and to both interpolate and extrapolate those results through time. Further, performance on the important and challenging task of agent forecasting is explored, and comparable performance to the state-of-the-art is achieved.

In future work the authors plan to extend this approach to the important task of multi-agent forecasting, where a normalizing flow formulation is expected to be particularly useful for capturing the complex high dimensional distributions.

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

## A. Related Works

The proposed framework is intrinsically related with two broad literature families: (1) ode based time-series forecasting models, and (2) distribution based forecasting models. In this section, we review statistical models from relevant literature, and discuss the difference between the proposed method with previous works.

**Neural ODEs for Time Series Forecasting.** Much recent work has explored embedding neural ODEs in models designed to process sequential data, like Recurrent Neural Networks (RNNs), replacing the hidden state with a neural ODE which evolves as a function of time (Rubanova et al., 2019; Brouwer et al., 2019; Voelker et al., 2019). These works are principally pre-occupied with solving the problem of encoding asynchronous time series data, in contrast we instead focus predicting the evolution of a probability distribution in what is assumed to be a continuous process.

Other recent works have used neural ODE based flows to connected multiple distributions (Li et al., 2020; Rempe et al., 2020). As in our architecture this models leverage a neural ODE flow to smoothly interpolate between multiple complex distributions. However unlike our model this transformation is not aligned with the temporal axis of the observed data. Similar to our proposed architecture, Tong et al. (2020) uses a neural ODE flow to connect predictions at several horizons, aligning ODE 'time' with the time of observations. However their model uses no conditional information, and generates plausible trajectories between observed data rather than attempting to forecast future marginal distributions.

In Deng et al. (2020) the model learns a distribution through time by flowing from the target distribution to a Wiener process. Similar to the work presented here this approach allows for an efficient estimation of the marginal distribution at any target horizon of interest. The key distinction is in their method the continuous prediction as a function of prediction horizon comes from the choice of a Wiener base distribution, separate from the choice of flow model. In our work the continuous behaviour is instead a direct result of the flow architecture used, defining a new temporal process rather than deforming an existing one.

Concurrent and closely related to our work is Chen et al. (2021), which explores a similar architecture for the related problem of point processes, and also utilizes a continuous normalizing flow to describe a marginal distribution across predicted event features as a function of target time. However their approach differs from our own as they are principally concerned with conditioning on the features and timing of past events, to predict the timing and features of discrete future events, where our model is concerned with the smoothly interpolated prediction of an underlying continuous process (e.g. the path of a vehicle) using a synthesis of extremely high dimensional conditioning information (lidar, cameras etc.). Practically this means that the way conditioning information is passed to the continuous flow model is quite distinct in the two approaches. Specifically in their model an attention mechanism allows sharp changes in the conditional distribution as a function of time, consistent with modelling a discontinuous point process. In our model a single vector of conditioning information in used across all time, consistent with modeling a continuous temporal process, and allowing for the smooth interpolation of marginals through time- a core functionality our model provides in contrast to other leading approaches.

**Distribution-Based Forecasting Models.** Autoregressive forecasting models provide a way to generate trajectories of any length(Whittle, 1951), with modern models allowing for the prediction of expressive distributions which in can capture complex multi modal behavior

(Salinas et al., 2019; Qiu et al., 2020) with a number of approaches utilizing normalizing flows in some way (Kumar et al., 2019; Shchur et al., 2019; Mehrasa et al., 2019; Bhattacharyya et al., 2019; Rasul et al., 2021) However in order to infer the statistics of a marginal distribution beyond the next time-step extensive sampling is required, and in these works a fixed discrete sampling in time is assumed.

Jain et al. (2019) proposes an architecture which, similar to our approach, explicitly relates marginal distributions in time. However their model is discrete in both time and agent position, and doesn't use the formalism of Normalizing Flows. Instead learning direct transforms on an discretized representation of the marginal distribution or an "occupancy grid" (Grigorescu et al., 2019; Mozaffari et al., 2019).

Rhinehart et al. (2019) describes a model which uses a series of affine transforms to learn a conditional joint distribution over a selection of agents and horizons. This formulation is similar to a discrete version of our model with a much less expressive choice of Normalizing Flow, and unlike our model is limited to only predict times seen in training.

Most similar to our model is Oh & Valois (2019) a conditional auto-regressive flow for marginal prediction at flexible horizons. Here however the flow model is a series of discrete layers, specifically a conditional extension of Neural Autoregressive Flows (Huang et al., 2018b) with the predicted horizon passed as an explicit conditioning variable.

# B. Method

Figure 3 shows an overview of our models computational graph.

## B.1. Mapping Data Time to ODE Time

One clear exception to the assumptions outlined earlier in the paper, is in the earliest possible predicted marginal, at some time $t$, with $t > t_0$. This distribution can be arbitrarily distinct from the base distribution. To solve this problem a "warm-up" period is introduced between the base distribution and the first evaluation point, with the length of the warm-up period optimized as a parameter in training. With this formulation, the translation from time in the target space $t_i$ to time in the ODE space $\tau_i$, given the warm-up period set by the parameter $\alpha$ is given simply as $\tau_i = \alpha + t_i$.

# C. Evaluation

## C.1. Position Estimation on Synthetic 2D Data

### C.1.1. SYNTHETIC GAUSSIANS

Following (Oh & Valois, 2019), we explore an extension of the synthetic Gaussian experiment from (Huang et al., 2018a), where a single model conditionally represents one

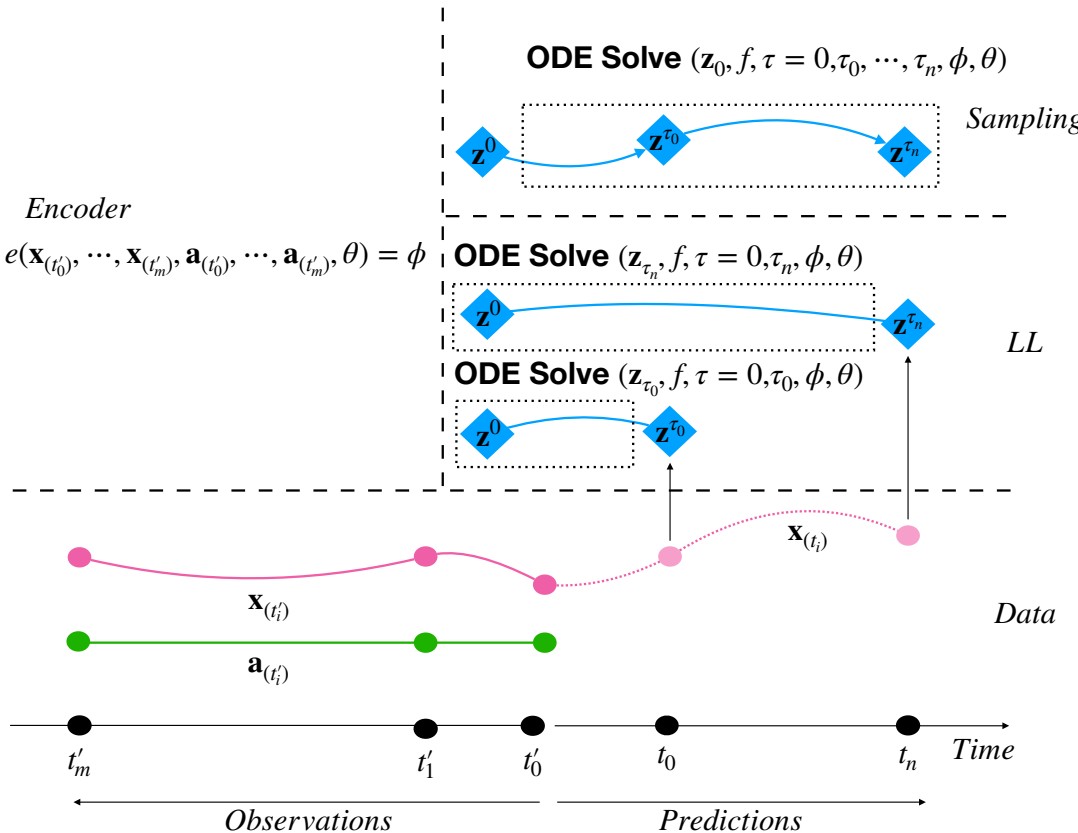

*Figure 3.* **Architecture.** Computation graph and model outline for our proposed architecture OMEN. **Data**. Shown in pink is the process we hope to predict, with observations $\mathbf{x}_{(t_i')}$ in the past and $\mathbf{x}_{(t_i)}$ in the future shown as circles. At inference only points $t_m'$ through $t_0'$ are available, with $t_0$ through $t_n$ used in training. The process shown in green represents additional conditioning information passed to the encoder that we don't intend to predict, reported at points $\mathbf{a}_{(t_i')}$ e.g. periodic lidar and video observations of the environment. **Encoder**. Observations from $t_m'$ through $t_0'$ are combined in a neural network to produce a single vector of conditioning information $\phi$. **LL**. Log-likelihood is calculated by solving our neural ode given the observation $\mathbf{z}_{\tau_n}$ at time ODE time $\tau_n$, and conditioning information $\phi$ to find the corresponding points in the base distribution $\mathbf{z}_0$ and the log determinant of the transform given by the trace of transform (boxed blue line). **Sampling**. Here we first sample from the base distribution to find $\mathbf{z}^0$, then solve for that point, conditioning information $\phi$, and n ODE time points of interest $\tau_0, \cdots, \tau_n$ to find points on the corresponding trajectory $\mathbf{z}^{\tau_0}, \cdots, \mathbf{z}^{\tau_n}$ (boxed blue line).

of three multi-modal configurations. For OMEN, conditioning information $n_m \in 0, 1, 2$ is encoded using an MLP before concatenated to every layer of the neural ODE flow in place of $\phi$. Results are shown in table 3, performance is comparable to the HCNAF approach and demonstrates that our choice of a conditional neural ODE based normalizing flow is capable of conditionally representing complex multi modal data.

| Model | AAF | NAF | HCNAF | OMEN |
|---|---|---|---|---|
| 2 by 2 | 6.056 | 3.775 | 3.896 | 3.896 |
| 5 by 5 | 5.289 | 3.865 | 3.966 | 3.975 |
| 10 by 10 | 5.087 | 4.176 | 4.278 | 4.336 |

*Table 3.* NLL for the synthetic Gaussian experiments. The AAF (Kingma et al., 2016) and NAF v results are for **individual models** for each configuration. The HCNAF and OMEN results are for a **single model** across all three configurations. Results for AAF, NAF, and HCNAF models are taken from (Oh & Valois, 2019).

### C.2. Precog Carla Dataset, Example Results

For Precog Carla dataset, an encoder network which is a partial re-implementation of that in (Oh & Valois, 2019), is used. LSTM modules encode the past trajectories of agents in the environment, and a residual CNN encodes Lidar information from a single main agent. Specifically two seconds of historical position data at a sampling of 5hz, or 10 historical points in time, are provided to the LSTM. The encoded trajectory and Lidar information is combined in a MLP and concatenated to every layer of a Neural ODE describing a normalizing flow, as outlined in Section 3.3. The model is trained and evaluated on the future position data of the main agent over four seconds at a sampling of 5hz, or 20 future time points.

In addition to Table 2, we also provide qualitative results. Figures 4, 5, and 6 show example predicted conditional marginal distributions for four of the twenty horizons in the Precog Carla Dataset. All examples are taken from the precog carla town01 test set.

#### C.2.1. BASELINES AND ABLATIONS

Minor extensions are made to the CTFP (Deng et al., 2020) model to provide a functional baseline. Specifically additional encoding information was concatenated with the output of the ODE-RNN, and an extra loss on extrapolating the predicted process into the future was added in training.

*OMEN-discrete* has a separate ODE flow transform between each inference time point in training. In this way it resembles a model following Eq. 1 where $\epsilon$ in the delta between forecast time points in the training set, and each neural ODE transform represents a separate but sequential normalizing flow transform. This ablation is expected to have *superior* expressive power as the representation no longer is con-

strained to be fully continuous in time, and each separate ODE transform can learn its own ODE stop time, allowing for expressive power between time steps to vary. This highlights a small degradation in performance from using a single Neural ODE to represent all time points, and suggests future approaches would benefit from a learned mapping of data time $t$ to ODE time $\tau$. However it does not allow for continuous interpolation of marginals in time.

*OMEN-nocon* has no conditioning information $\phi$ appended to the neural ODE. This ablation is expected to have significantly worse overall performance as the model only learns a distribution over all points observed in the training set, and we expect the task of predicting agent locations to be strongly conditional on the available environmental information. This highlights the importance the extension to (Chen et al., 2018; Grathwohl et al., 2018) presented in this paper to include conditioning information.

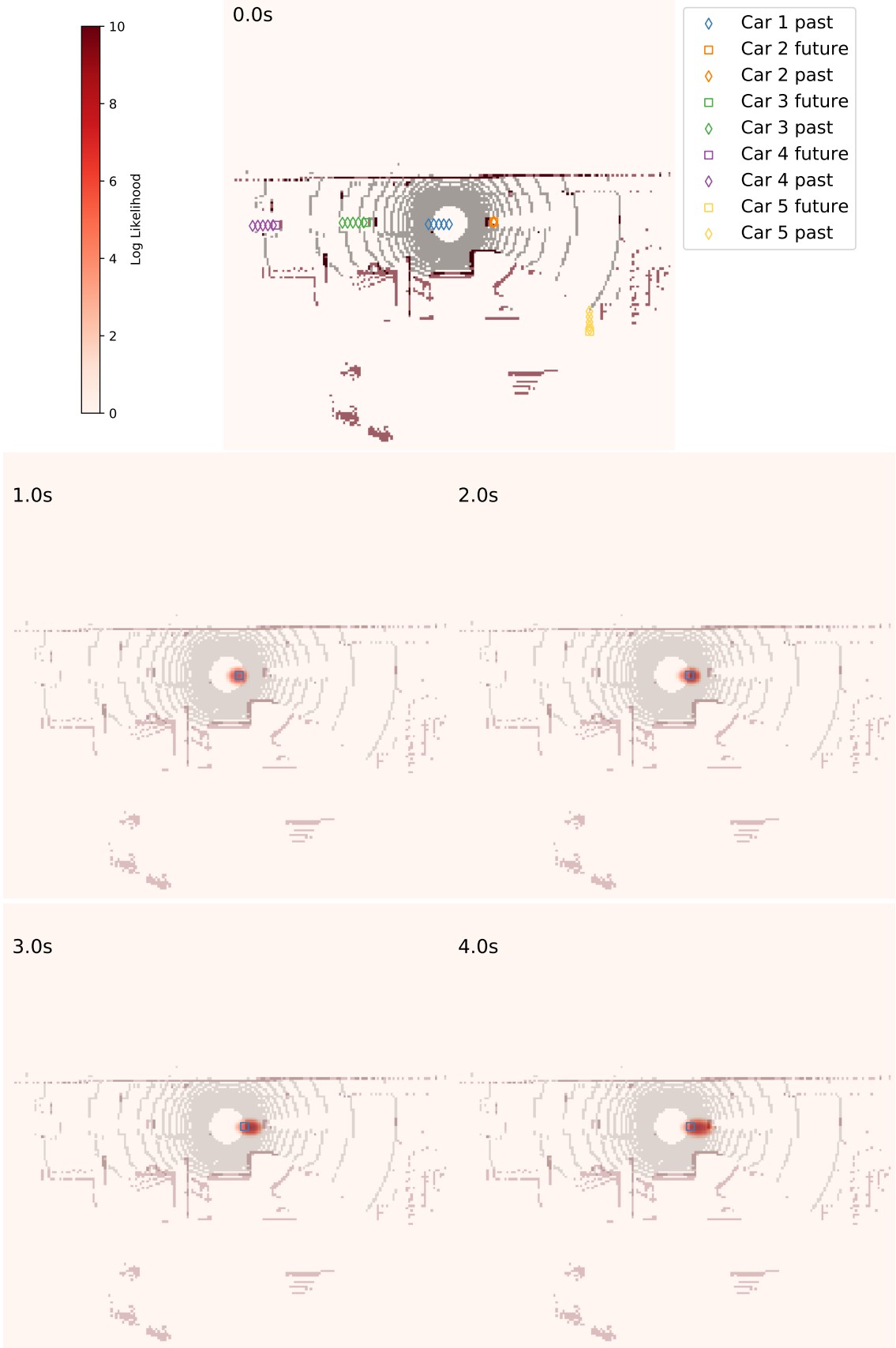

*Figure 4.* **Example Preco-Carla Prediction** example predicted conditional marginal distributions for four of the twenty horizons in the Precog Carla Dataset. The full conditioning information available to the agent is shown at the top, specifically the autopilots historical trajectory, the historical trajectory of the four closest cars, and a lidar captured by the autopilot at $t = 0$. A single future point for each agent is appended to the top plot to aid the reader when estimating the direction of those agents. The four bottom plots show marginals at $t \in 1, 2, 3, 4s$ into the future and the true future location of the autopilot at those times.

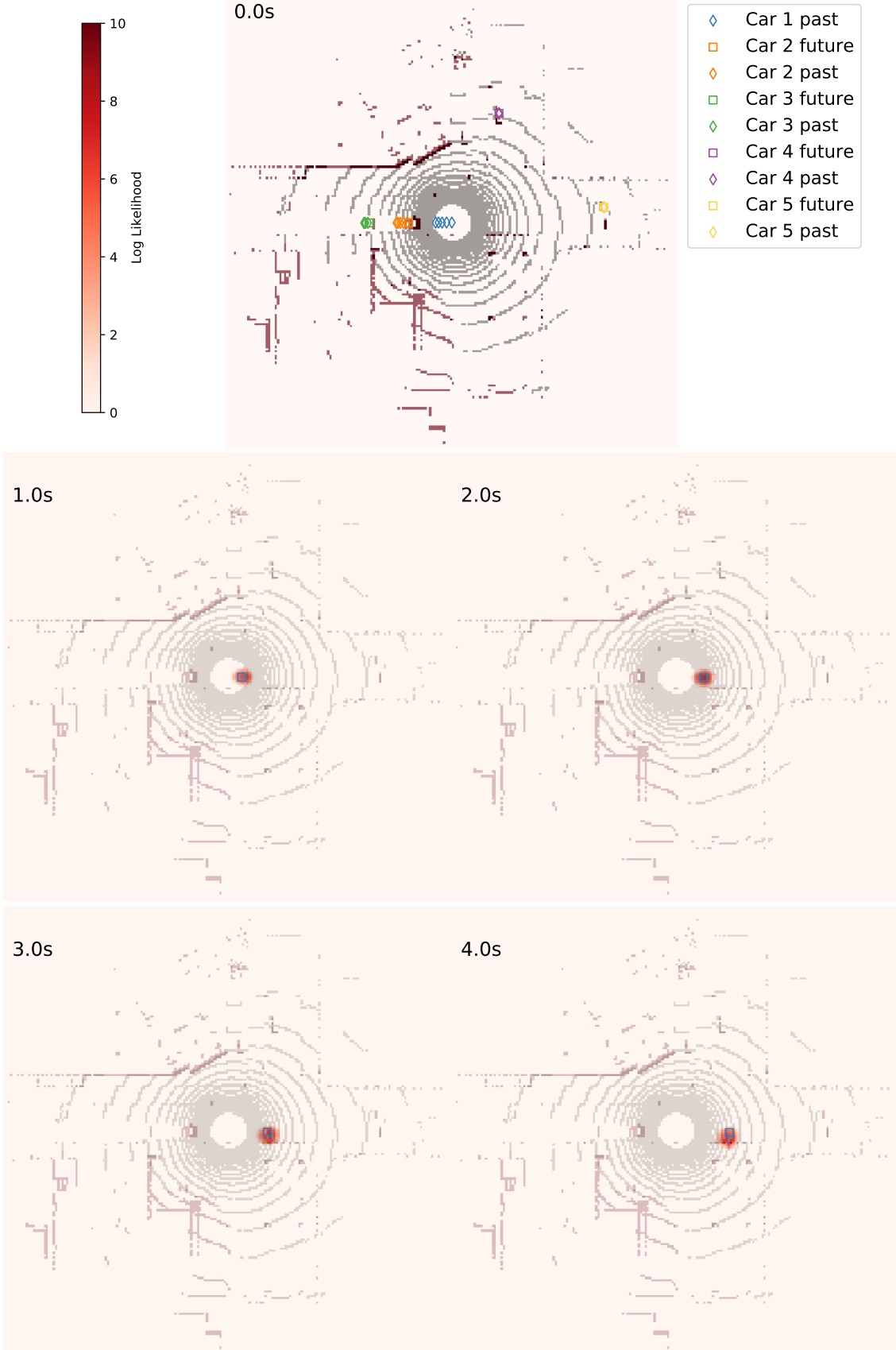

*Figure 5.* **Example Preco-Carla Prediction** example predicted conditional marginal distributions for four of the twenty horizons in the Precog Carla Dataset. The full conditioning information available to the agent is shown at the top, specifically the autopilots historical trajectory, the historical trajectory of the four closest cars, and a lidar captured by the autopilot at $t = 0$. A single future point for each agent is appended to the top plot to aid the reader when estimating the direction of those agents. The four bottom plots show marginals at $t \in 1, 2, 3, 4s$ into the future and the true future location of the autopilot at those times.

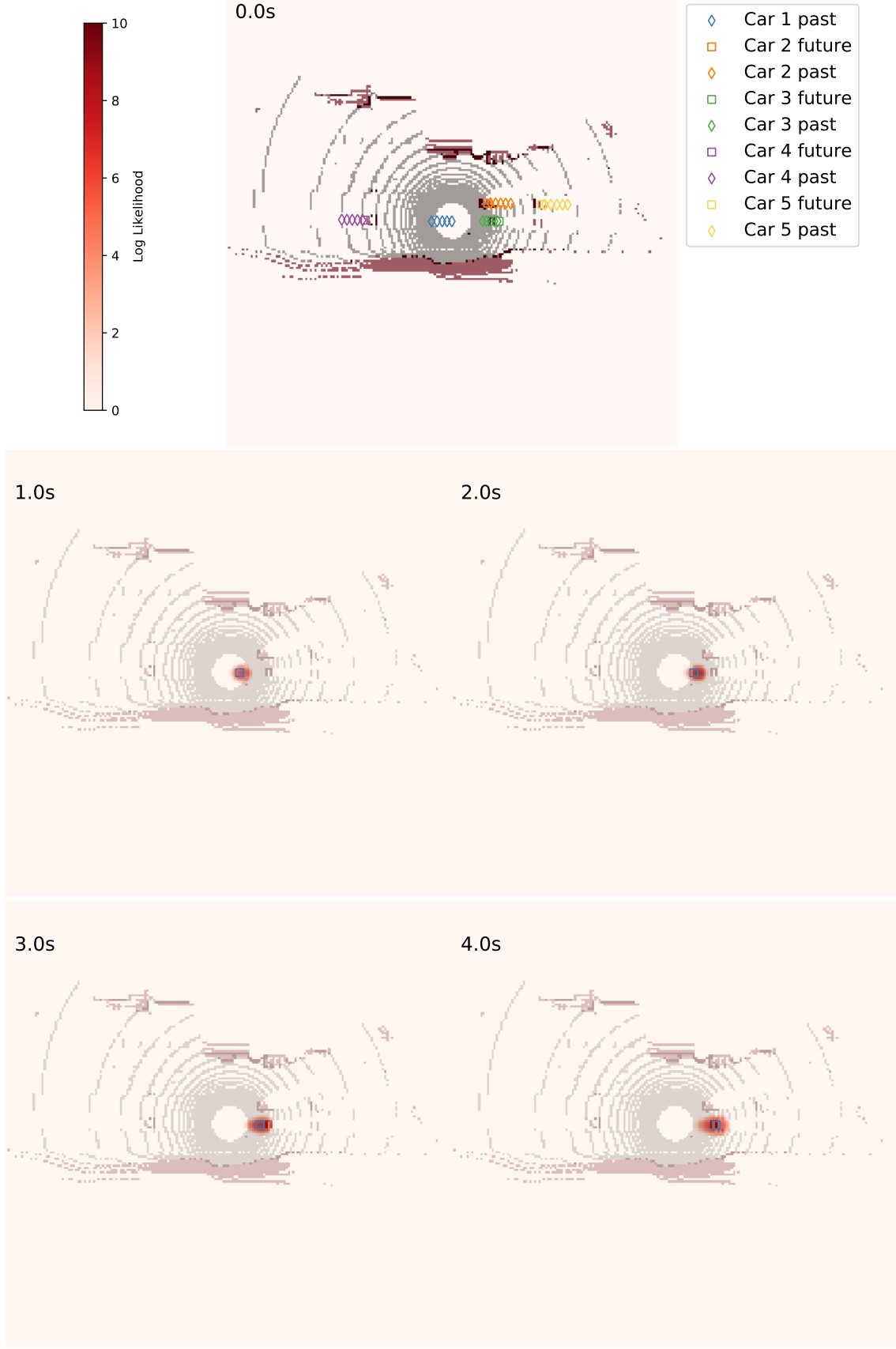

*Figure 6.* **Example Preco-Carla Prediction** example predicted conditional marginal distributions for four of the twenty horizons in the Precog Carla Dataset. The full conditioning information available to the agent is shown at the top, specifically the autopilots historical trajectory, the historical trajectory of the four closest cars, and a lidar captured by the autopilot at $t = 0$. A single future point for each agent is appended to the top plot to aid the reader when estimating the direction of those agents. The four bottom plots show marginals at $t \in 1, 2, 3, 4s$ into the future and the true future location of the autopilot at those times.