# OpenReview forum: "Agent Forecasting at Flexible Horizons using ODE Flows"
_ICML.cc/2021/Workshop/INNF — INNF+ 2021 poster_

### Official Review · Reviewer_iCJp · 2021-06-10

**Rating:** Borderline Accept
**Confidence:** 3

**Summary:**

This paper proposed a method to forecast the marginal positional distributions of motion trajectories using a flow constructed where the temporally-preceding output distribution is used as the base distribution for the next prediction (Eq 1); this discrete version seems to be discussed in order to motivate the construction of the proposed conditional continuous-time neural ODE model -- it's not quite clear why the "discrete" version of the proposed model is introduced, mainly because the connection between it and the model described in Eq (2) is not clear. The paper presents experiments on both synthetic and real datasets of vehicle motion.

A clearer description of the notation at the beginning of the Method section would improve the paper. It's not clear what the single-quote character is being used to denote. Maybe use \mathcal to denote sets (e.g. for T, T', A'?)

A clear description of the challenge that asynchronous arrival of agents would improve the paper. From L83 it looks like the model considers only pairs of agents coupled together (i.e. doesn't consider modeling higher-order relationships)?

L148-L164. It's not clear what exactly is extended from Chen et al. 2018 ; Grathwohl et al. 2018 (it seems to be just the conditioning?). The paper should be more explicit about the differences. Perhaps a comparison to these approaches is also warranted (e.g. would they learn an unconditioned prior, which should be expected to perform worse in the conditional setting?)

L150 It's not clear what the vector of encoding information is (z? or \phi? L196 suggests \phi but L150 needs to be clear about this).

Single-agent prediction is evaluated, yet the method seemed to propose a multi-agent model. The paper should be clearer about which domain it focuses on.

More clarity is needed on the evaluation -- the paper discusses estimating the _marginal_ distributions of future timesteps, yet the e-hat metric from PRECOG evaluates the likelihood of trajectories under the joint distribution. I think there's a type mismatch here. The paper should clarify this issue and potentially change its evaluation procedure if there is indeed a type mismatch.

More discussion of how the approach differs from Deng et al. 2020 is needed.


**Justification For Rating:**

The method seems promising, but the writing is not as clear as it could be, and I have some doubts about part of the experimental evaluation

---

### Official Review · Reviewer_m773 · 2021-06-11

**Rating:** Borderline Accept
**Confidence:** 3

**Summary:**

This paper applies continuous normalizing flows (i.e. neural ODE) to the problem of predicting the distribution of possible positions of agents at arbitrary times in the future, conditioned on asynchronous observations in the past. By using the predicted marginal distribution at some point in time as the reference distribution for the next flow "step", an assumption about smoothness of those predictions over time is encoded and enables smooth interpolation. The method is demonstrated on a relevant benchmark and its special properties experimentally verified on suitable toy data.

**Justification For Rating:**

The concept makes sense to me and is communicated well in the paper. It is difficult for me to judge the significance of the experiments, having no background in agent forecasting literature and challenges. It seems to me that the method works as intended, without representing a radical improvement over the field.

My main conceptual doubt is this:
The same network is used to (a) transforms a Gaussian latent distribution to the prediction at the first time point, and (b) transform predictions at various time points to similar distributions at later times. Aren't (a) and (b) very different types of transport, since the initial Gaussian should have a very different shape from the typical prediction distributions that follow along the flow? I wonder, if I got this right, how the network differentiates between these two "regimes" and if this takes a toll on its capacity to solve the task.

---

### Official Review · Reviewer_9fdw · 2021-06-12

**Rating:** Borderline Accept
**Confidence:** 3

**Summary:**

This work proposes a conditional continuous normalizing flow to predict density evolution in time. The proposed model is essentially a continuous normalizing flow with a neural ODE conditioned on a context vector given by an encoder neural network. Authors demonstrate that their model performs decently on 2D synthetic data time series and on the benchmark comprised of simulated trajectories of an autopilot (PRECOG Carla dataset).

Overall:
* It is hard to understand the architecture from the text due to confusing notations and a lack of details
* The evaluation looks promising and results are interesting for community
Detailed comments:
1. Figure 3 in the appendix helps to understand overall structure of the model but notations there are very confusing. Firstly, t’ and t notations for time points are confused between the body of the paper and the appendix. In the body t’ denotes input observation time points and t denotes target time points while in the appendix these notations are inverted. Secondly, It is not clear how vector z is connected with target predictions and how time points denoted as tau correspond to target time points t’. Finally, the same notation f is used to denote the encoder network and the neural ode network which is confusing.
2. It would be easier to understand the architecture if authors added more detail about what the input and the output of the model are. Especially, it’s worth to mention the length of the input sequence of observations. Moreover, it is not clear whether numerical integration is performed separately for each target time point (meaning the predictions for different target time points are not directly dependent) or the sequence of predictions was obtained by one forward pass of an ODE solver (meaning that predictions are obtained sequentially and the current prediction depends a lot on the previous one).
3. The comparison with OMEN-discrete version of the model is posed as an ablation study. However, it is not clear what this comparison shows to the reader. OMEN-discrete is more flexible than OMEN as it consists of several sequential continuous normalizing flows, so it works expectedly better. Comparison with such an ablation neither highlights importance/benefits of a particular component of the proposed model nor shows its limitations. I would recommend authors to add comparison with simplified versions of the OMEN (e.g. excluding conditioning to show its importance) and make conclusions from the comparison with OMEN-discrete more clear.

**Justification For Rating:**

The proposed model and results look interesting for community although they are not described clear enough

---

### Decision · Program_Chairs · 2021-06-14

**Decision:**

Accept (poster)

**Comment:**

The paper is on topic for this workshop and the reviewers were (moderately) positive. We have therefore accepted this paper to the workshop. Please take into account the reviewers' suggestions when submitting the camera ready version.